# Plasmidome of *Salmonella enterica* serovar Infantis recovered from surface waters in a major agricultural region for leafy greens in California

**Beatriz Quiñones**, **Bertram G. Lee, Ashley Avilés Noriega, Lisa Gorski** *

Produce Safety and Microbiology Unit, Western Regional Research Center, Agricultural Research Service, U.S. Department of Agriculture, Albany, California, United States of America

* Lisa.Gorski@usda.gov

**Data Availability Statement:** The annotated nucleotide sequences of the 14 plasmids in the examined S. enterica serovar Infantis strains are found under the GenBank accession numbers

## Abstract

Non-typhoidal *Salmonella enterica* is a leading cause of gastrointestinal illnesses in the United States. Among the 2,600 different *S. enterica* serovars, Infantis has been significantly linked to human illnesses and is frequently recovered from broilers and chicken parts in the U.S. A key virulence determinant in serovar Infantis is the presence of the megaplasmid pESI, conferring multidrug resistance. To further characterize the virulence potential of this serovar, the present study identified the types of plasmids harbored by Infantis strains, recovered from surface waters adjacent to leafy greens farms in California. Sequencing analysis showed that each of the examined 12 Infantis strains had a large plasmid ranging in size from 78 kb to 125 kb. In addition, a second 4-kb plasmid was detected in two strains. Plasmid nucleotide queries did not identify the emerging megaplasmid pESI in the examined Infantis strains; however, the detected plasmids each had similarity to a plasmid sequence already cataloged in the nucleotide databases. Subsequent comparative analyses, based on gene presence or absence, divided the detected plasmids into five distinct clusters, and the phylogram revealed these Infantis plasmids were clustered based either on the plasmid conjugation system, IncI and IncF, or on the presence of plasmid phage genes. Assignment of the putative genes to functional categories revealed that the large plasmids contained genes implicated in cell cycle control and division, replication and recombination and defense mechanisms. Further analysis of the mobilome, including prophages and transposons, demonstrated the presence of genes implicated in the release of the bactericidal peptide microcin in the IncF plasmids and identified a Tn10 transposon conferring tetracycline resistance in one of the IncI1 plasmids. These findings indicated that the plasmids in the environmental *S. enterica* serovar Infantis strains from surface waters harbored a wide variety of genes associated with adaptation, survivability and antimicrobial resistance.

PQ365737-PQ365750 (Table 1) in the National Center for Biotechnology Information (NCBI) nucleotide database (https://www.ncbi.nlm.nih.gov/nuccore/), associated with the BioProject accession number PRJNA1162552.

**Funding:** This work supported by the United States Department of Agriculture, Agricultural Research Service, CRIS Project Numbers 2030-42000-052-00D and 2030-42000-055-00D. The funders had no role in study design, data collection and analysis, decision to publish, or preparation of the manuscript. Mention of trade names or commercial products in this publication is solely for the purpose of providing specific information and does not imply recommendation or endorsement by the USDA. USDA is an equal opportunity provider and employer.

**Competing interests:** The authors have declared that no competing interests exist.

## Introduction

Non-typhoidal *Salmonella enterica* is one of the leading causes of foodborne illness in the United States [1–3]. Salmonellosis is a gastrointestinal illness of mild to moderate severity that can manifest approximately 72 h after exposure to infected food [3, 4]. The species *S. enterica* contains over 2,600 different serovars, which are differentiated by the combinations of the various somatic O and two phases of flagellar H antigens [5]. However, only about 100 serovars are implicated in most of the illness in humans [6, 7]. Since 2001, there has been a 167% increase in the incidence rate in infections caused by *S. enterica* serovar Infantis (*S.* Infantis) [6, 8]. Among the serovars causing human illness, incidence of illness due to *S.* Infantis moved from the ninth most common in 1996–1998 to the sixth most common in 2019 [9] and since 2016, this serovar has been among the top five isolated from broilers and chicken parts [10]. One factor recognized in the emergence of serovar Infantis strains is the acquisition of a conjugative mega plasmid, termed as plasmid of emerging *S. enterica* Infantis (pESI) [8, 11, 12]. First described in isolates from Israel in 2014, plasmid pESI was found to harbor multiple antimicrobial resistance genes, including an extended-spectrum-β-lactamase (ESBL) gene [11]. Since 2014, serovar Infantis isolates with this plasmid have been isolated worldwide, primarily from raw poultry; however, it is also present in wildlife and the environment [8, 12, 13]. Multiple drug resistant strains of *S.* Infantis were reported as early as 1979 in isolates from wild birds in Florida [14].

Plasmids are vectors for transferring resistance, virulence, and adaptation genes in bacteria [15–17]. As mediators of horizontal gene exchange, plasmids carry various types of resistance genes, including antibiotic resistance, metal resistance, and biocide resistance. They also can provide rapid adaptation to an organism during changing environmental conditions [15–17]. In the environment, strains of *Salmonella* are adept at cycling between host and non-host environments, including surface water, and can tolerate stress and nutrient depletion [18–23]. In several surveys of an important produce-growing region along the Central California coast, *Salmonella* was routinely isolated, and some strains and subtypes showed indications of persistence and/or regular re-introduction into the environment [24–27]. In one survey of *Salmonella* from amphibians, reptiles, and surface waters, approximately 62% of the 106 *Salmonella* strains isolated were resistant to at least one antibiotic [26].

During a 5-year longitudinal survey (2011–2016) of surface waters of the Central California Coastal region, strains of *S.* Infantis were isolated from 34 samples collected during four of the five years from 13 of the 30 regular sampling sites [27]. Because of the increasing visibility of serovar Infantis strains, and the importance of plasmids in that emergence, the *S.* Infantis strains from that water survey were studied in depth for the occurrence of plasmids. The aim of this study was to determine plasmid carriage in the strains, and to characterize the types of plasmids among *S.* Infantis strains in this California water environment.

## Materials and methods

### Sampling, bacterial culturing, and DNA extraction

The environmental *S.* Infantis strains used in this study were isolated from surface waters collected during the sampling years 2012 to 2016 [27] (S1 Table). All of the sampling sites were located on public lands within an approximate 20-mile radius of Salinas, California (USA); therefore, no specific permissions were required for the public lands sampling, which did not involve any endangered or protected species. Genomic DNA was extracted from single colonies grown in Trypticase soy broth + 0.6% yeast extract (TSYE, both ingredients from Difco, Franklin Lakes, Nj) using the Wizard Genomic DNA purification kit (Promega, Madison, WI)

according to manufacturer's instructions except that the DNA was resuspended in Buffer EB (Qiagen, Hilden, Germany). Plasmids were extracted as described [28]. Briefly, *Salmonella* isolates were streaked on TSA agar and incubated overnight at 37˚C. Several large colonies were smeared in microcentrifuge tubes and resuspended in lysis buffer followed by incubation at 55˚C for 60–90 minutes. Cell lysates were extracted with 100 µl phenol:chloroform:isoamyl alcohol (25:24:1) from which 50–70 µL of DNA were recovered. Samples were electrophoresed on a 0.7–0.8% agarose gel stained with GelRed in 1X Tris Borate EDTA buffer at 4˚C for 5 hours at 100V and visualized in a UV imager. Tetracycline resistance was assessed using the Kirby-Bauer method [29] with the BD BBL™ Sensi-Disc™, impregnated with 30 µg of tetracycline (Becton-Dickinson, Franklin Lakes, NJ). Mueller-Hinton agar (Difco) was swabbed with bacterial cultures grown in TSB to an early logarithmic phase. The disks were deposited onto the bacterial lawn, and the plates were further incubated for 18 hours at 37˚C. After incubation, the inhibition zones were measured, and the resistance/susceptibility was then determined by following the antimicrobial susceptibility testing guidelines [30].*Escherichia coli* ATCC strain 25922 was a control for the antibiotic resistance assay.

## Library preparation and sequencing

Genomic DNA was prepared using the Wizard Genomic DNA Purification Kit (Promega, Madison, WI). The concentration and quality of the genomic DNA extracted from the recovered *Salmonella* strains was determined by using a NanoDrop ND-1000 spectrophotometer and a Qubit 4 fluorometer (Thermo Fisher Scientific, Waltham, MA USA). Long-read genome sequencing was performed using a MinION device (Oxford Nanopore Technologies, Oxford, UK). The sequencing libraries were prepared using a total of 200 ng DNA per *Salmonella* strain using a Rapid Barcoding Sequencing kit SQK-RBK-114.24 (Oxford Nanopore Technologies), following the manufacturer's workflow recommendations and procedures (https://community.nanoporetech.com/docs/prepare/library_prep_protocols/rapid-sequencing-gdna-barcoding-sqk-rbk114/v/rbk_9176_v114_revm_27nov2022/library-preparation?devices=minion (accessed on 1 March 2024). The prepared libraries were subsequently sequenced on FLO-MIN114 (version 10.4.1 active pore number ≥1100) flow cells (Oxford Nanopore Technologies) for a total of 48 h using MinKNOW software (v24.02.10) with the depletion option (adaptive sampling) using the reference genome from *S*. Infantis strain CVM N18S1246 (GenBank accession no. CP052777.1) to selectively increase the proportion of plasmid DNA to genomic DNA sequenced [31]. The sequencing reads were base called, trimmed, demultiplexed on super high accuracy mode, filtered on Phred Quality score 8, and saved in the FASTQ format. Subsequently, the error correction and assembly of the sequencing reads were performed with Flye assembler version 2.9 [32] by using the recommended preset options for sequencing reads obtained with the Oxford Nanopore Technologies platform.

To resolve the plasmid sequences in the *S*. Infantis strains RM18512 and RM18281 into a single contig, the genomes for these strains were sequenced using the Illumina MiSeq System with 500-cycle v2 paired-end sequencing kits (Illumina, San Diego, CA, USA), following the manufacturer's instructions. DNA libraries were prepared and quantified using the KAPA Low-Throughput Library Preparation Kit and KAPA Library Quantification Kit (Roche, Kapa Biosystems, Wilmington, MA) with modifications as previously described [33]. Libraries were validated with the Agilent 2100 Bioanalyzer chips using the Agilent 12000 kit (Agilent, Santa Clara, CA). The combination of MiSeq and Nanopore sequencing reads were combined by using Unicycler software version v0.4.8 to obtain accurate and complete assemblies of short and long reads [34].

### Sequence annotation and comparative analyses

The annotation of the plasmid sequences was performed by using RAST version 2.0 pipeline [11], were visualized using SnapGene software version 7.2 (GSL Biotech LLC, San Diego, CA), and were converted to a five-column tab-delimited annotation file using the GB2Sequin web application prior to submission to GenBank [35]. Confirmation of the Infantis serovar in the examined strains was performed with the SeqSero2 algorithm [36]. To identify the conjugation systems and mobilization families for each plasmid, the amino acid sequences were uploaded to the PlasmidFinder service to determine the mobilization family with MOBscan and to the Galaxy ToolShed site for identifying the conjugation system type with CONJscan [37–40]. To screen the plasmid sequences, the Virulence Finder Database and the Comprehensive Antibiotic Resistance Database [41, 42] were employed to search for genes related to virulence and antimicrobial resistance traits, respectively, and both databases were used in conjunction with the ABRicate tool, version 1.0.1, by selecting the parameter cutoffs of 90% coverage and 95% nucleotide identity [43]. Plasmids also were analyzed with PHASTEST, a web server designed to identify prophage sequences [44]. Furthermore, the comparative analysis of the plasmid sequences was conducted using the Anvi'o platform [45], version 7, by selecting the pangenomics workflow system. The function in the Anvi'o workflow called "anvi-compute-gene-cluster-homogeneity" was employed to compute the geometric and functional homogeneity indexes, and these indexes provided measures of similarity in the detected plasmid genes with an index of 1 being the highest value to indicate no variation. The results obtained using the annotated genomic databases were then subsequently visualized using the command entitled "Anvi-display-pan". Additionally, the data in the gene cluster summary file from the Anvi'o workflow system were used to analyze the distribution of functional clusters of orthologous gene categories in the examined *Salmonella* strains. The annotated nucleotide sequences of the 14 plasmids in the examined *S. enterica* serovar Infantis strains are found under the GenBank accession numbers PQ365737-PQ365750 (Table 1) in the National Center for Biotechnology Information (NCBI) nucleotide database (https://www.ncbi.nlm.nih.gov/nuccore/), associated with the BioProject accession number PRJNA1162552.

## Results

### Comparative analysis of the non-pESI plasmids in the environmental *S. enterica* serovar Infantis strains

Initial sequencing analysis showed that 14 strains had a plasmid with sizes ranging from 78 kb to 125 kb (Table 1). The remaining 20 *S.* Infantis strains did not show bands in the plasmid screening (S1 Table). Twelve strains were selected for subsequent analysis due to high yield recovery of the plasmid DNA. BLASTN searches did not show sequence similarity to the emerging conjugative mega plasmid, pESI; however, each plasmid sequence had some similarity to a plasmid already cataloged in the database (Table 1). A comparative gene analysis was then conducted by employing the Anvi'o workflow to subsequently characterize the plasmid sequences in the examined *S.* Infantis strains. Phylogenetic analysis, based on gene presence or absence, in the *S.* Infantis strains divided the plasmid sequences into five distinct grouping (clusters) (Fig 1), and the phylogram revealed that the examined strains were clustered based on the plasmid conjugation system.

Moreover, further analysis of the contigs revealed a small multicopy 4 kb-plasmid in strains RM17050 and RM16752, which was assigned to cluster I, and displayed 100% identity (99% coverage) to plasmid pST46-2 from *S. enterica* serovar Typhimurium (Fig 1 and Table 1). The IncI1-based conjugation system, belonging to the MOB$_P$ family, was identified in cluster II

**Table 1. Plasmid characteristics in the examined *S. enterica* serovar Infantis strains from surface waters.**

| Cluster | *S.* Infantis strain ID | Plasmid name | Size (bp) | MOB family[a] | Conjugation system[b] | Number of genes[c] | Number of functional categories[d] | GenBank accession number | GenBank accession number of related plasmid (plasmid name) |
|---|---|---|---|---|---|---|---|---|---|
| I | RM16752 | pRM16752-2 | 4,074 | none | none | 4 | 2 | PQ365738 | CP050752.1 (plasmid pST46-2) |
| | RM17050 | pRM17050-2 | 4,074 | none | none | 4 | 2 | PQ365739 | CP050752.1 (plasmid pST46-2) |
| II | RM17050 | pRM17050-1 | 78,346 | MOB$_P$ | IncI1 | 85 | 19 | PQ365740 | CP136836.1 (plasmid pK2721-1) |
| | RM18148 | pRM18148 | 91441 | MOB$_P$ | IncI1 | 103 | 25 | PQ365744 | OP378609 (plasmid pEC00592_5) |
| | RM18410 | pRM18410 | 88,744 | MOB$_P$ | IncI1 | 102 | 25 | PQ365746 | LC567088 (plasmid pCH56) |
| | RM16752 | pRM16752-1 | 84,544 | MOB$_P$ | IncI1 | 98 | 21 | PQ365737 | CP136836.1 (plasmid pK2721-1) |
| | RM18147 | pRM18147 | 84,109 | MOB$_P$ | IncI1 | 99 | 23 | PQ365743 | AP021916 (plasmid pWP2-W18-CRE-03_2) |
| III | RM21154 | pRM21154 | 90,009 | None | None | 119 | 24 | PQ365750 | CP111090 (plasmid p2013K-1747) |
| | RM17884 | pRM17884 | 92,091 | None | None | 115 | 23 | PQ365741 | CP111090 (plasmid p2013K-1747) |
| | RM17885 | pRM17885 | 87,819 | None | None | 121 | 25 | PQ365742 | CP111090 (plasmid p2013K-1747) |
| | RM21071 | pRM21071 | 87,783 | None | None | 113 | 23 | PQ365748 | CP111090 (plasmid p2013K-1747) |
| IV | RM21082 | pRM21082 | 91,487 | MOB$_F$ | IncF | 108 | 40 | PQ365749 | CP077697 (plasmid pCFSAN058598_01) |
| V | RM18512 | pRM18512 | 116,460 | MOB$_F$ | IncF | 142 | 58 | PQ365747 | CP022660 (plasmid pRM11060-2) |
| | RM18281 | pRM18281 | 126,613 | MOB$_F$ | IncF | 158 | 58 | PQ365745 | CP022660 (plasmid pRM11060-2) |

[a]The mobilization (MOB) family was determined with MOBscan by uploading the amino acid sequences to the PlasmidFinder service.

[b]The conjugation system was determined with CONJscan by uploading the amino acid sequences to the Galaxy ToolShed site.

[c]Number of genes represents coding sequences as identified using RAST version 2.0 pipeline.

[d]Number of functional categories represents the clusters of orthologous genes, based on the gene cluster summary file with the Anvi'o workflow.

plasmids pRM16752-1, pRM17050-1, pRM18147, pRM18148 and pRM18410. Plasmids pRM17050-1 (86% coverage) and pRM16752-1 (99% coverage) showed 95–99% similarity, respectively, to pK2721-1 from *Citrobacter koseri* strain (Table 1). A high similarity of 99% was also observed for plasmids pRM18148 and pRM18410 with the clinical *Escherichia coli* plasmid pEC00592_5 (100% coverage) and the chicken meat *E. coli* plasmid pCH56 (94% coverage), respectively. Cluster II plasmid pRM18147 showed 98% similarity (96% coverage) with plasmid pWP2-W18-CRE-03_2 from *E. coli* (Table 1), recovered from wastewater.

The IncF-based conjugation system, belonging to the MOB$_F$ family, was detected in plasmids pRM18512, pRM18281 and pRM21082. However, cluster IV plasmid pRM21082, displaying 100% similarity (96% coverage) to plasmid pCFSAN058598_01, was only identified in strain RM21082 and was found to be 3000 bp smaller than the plasmids pRM18512 and pRM18281, which were assigned to cluster V and showed sequence similarity to plasmid pRM11060-2 (100% identity; 40% coverage) (Table 1). Moreover, plasmids pRM17884, pRM17885, pRM21071 and pRM21154, assigned to cluster III, did not contain a conjugation system (Table 1); however, further analysis of the sequences with PHASTEST revealed that the cluster III plasmids resembled a phage plasmid, and showed the closest sequence similarity

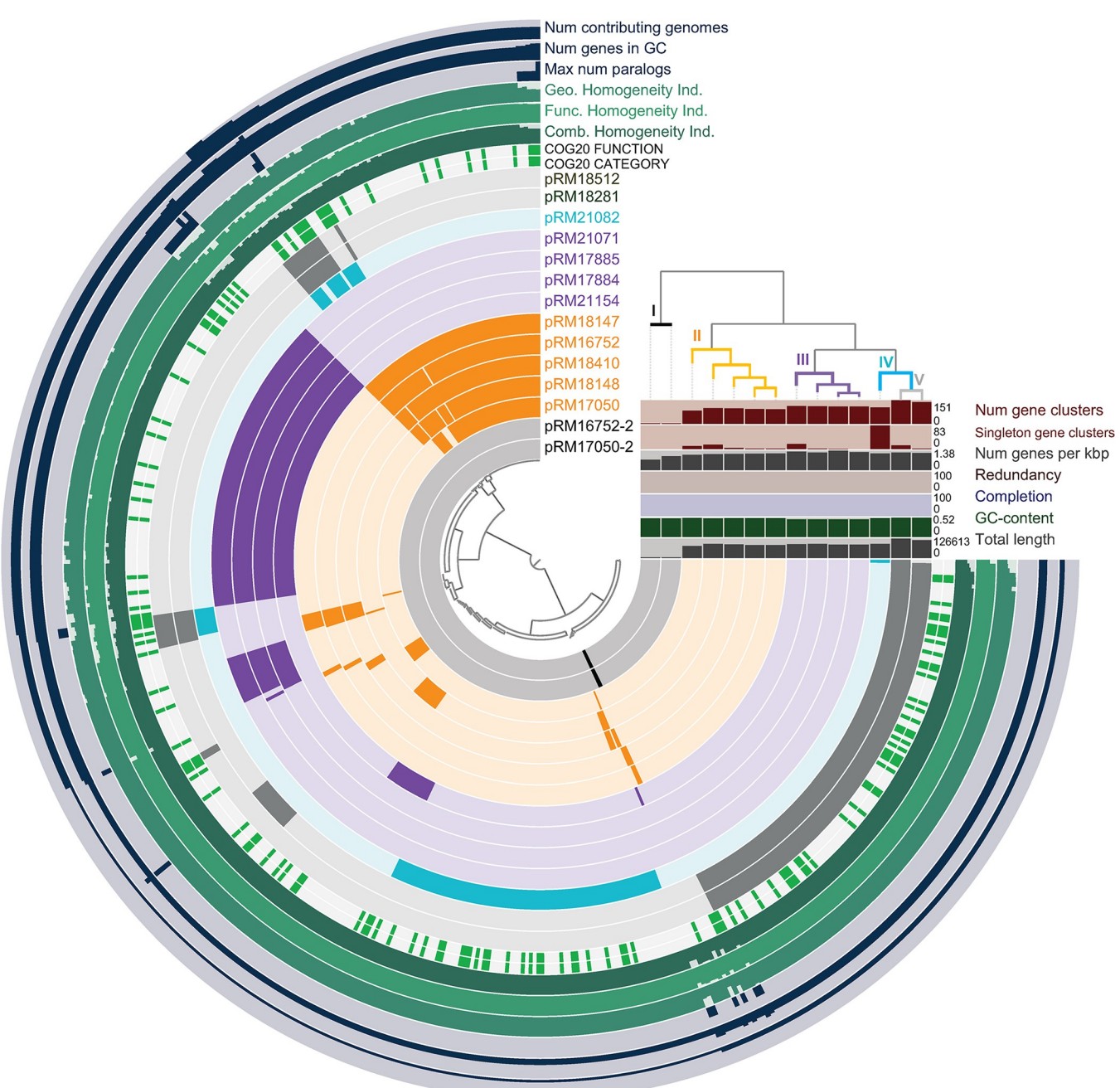

**Fig 1. Comparative sequence analysis of the examined plasmids from *S. enterica* Infantis.** The plasmid sequences identified *S. enterica* Infantis strains were annotated and further compared with the Anvi'o pangenome workflow and visualized with the "Anvi-display-pan" command.

(99% identity; 88–99% coverage) to a plasmid found in the clinical strain, *S. enterica* 2013K-1747 (Table 1). Plasmid pRM18418 showed 99% sequence similarity to plasmid pN18S0935-1 (98% coverage) in *S.* serovar Kentucky strain N18S0935 and to plasmid pN17S166 (92% coverage) recovered from *S. enterica* serovar 4, [5],12:i:− strain CVM N17S166.

As further demonstrated in the comparative analysis with the Anvi'o workflow (Fig 1), various homogeneity indexes were determined to measure the variation and frequency of their occurrence in the detected plasmid genes with an index of 1 being the highest value indicating

no variation. In this study, the average homogeneity index value was determined to be 0.95 to 0.96 for all plasmid sequences combined. Subsequent measurement of the geometric and functional homogeneity indexes was performed. In particular, the geometric homogeneity index determined the similarity between gene variations based on how many gaps exist between the different sequences in the gene. The functional homogeneity index measured the similarity between gene variations based on how many differences exist between aligned sequences of the gene variations. The results obtained in this study identified notable genes, linked to plasmid stability, segregation and replication, with lower geometric homogeneity such as the DNA methylase subunit (HsdM) in the IncI1 plasmids with an index of 0.64, the chromosome segregation protein (Spo0J) in the IncF plasmids with an index of 0.65, and the DNA polymerase (DinB) with an index of 0.64. Additionally, other identified genes, associated with plasmid transferability and replication, with lower functional homogeneity were the single-stranded DNA-binding protein (Ssb) among the IncF and IncI1 plasmids with an index of 0.80, the type IV secretory pathway component (VirB4) among the IncF plasmids with an index of 0.71, and the SOS-response transcriptional repressor (LexA) among the IncF and IncI1 plasmids with an index of 0.81.

## Characterization of the functional gene categories in the examined plasmids from *S. enterica* serovar Infantis strains

Further analysis of the detected genes in the Infantis plasmids revealed that on average 25% of the genes in all of the examined plasmids were assigned to functional categories (Table 1). In particular, the two small 4 kb-plasmids in cluster I had the highest proportion of assigned categories. The cluster II and cluster III plasmids with an average plasmid size of 85.4 kb to 89.4 kb had about 20–23% of detected genes with an assigned function. The largest plasmids in clusters IV and V with an average size of 91.5 kb and 121 kb, respectively, were found to have a higher proportion of the genes (38%) assigned to functional categories (Table 1). By conducting a subsequent analysis with the Anvi'o workflow, the presence of putative genes for strains belonging to each grouping was identified, and the genes assigned to specific functional categories were further examined (Fig 2). All the large plasmids (>78 Kb) contained at least 4% of genes in the functional category for cell cycle control and cell division (D category), replication

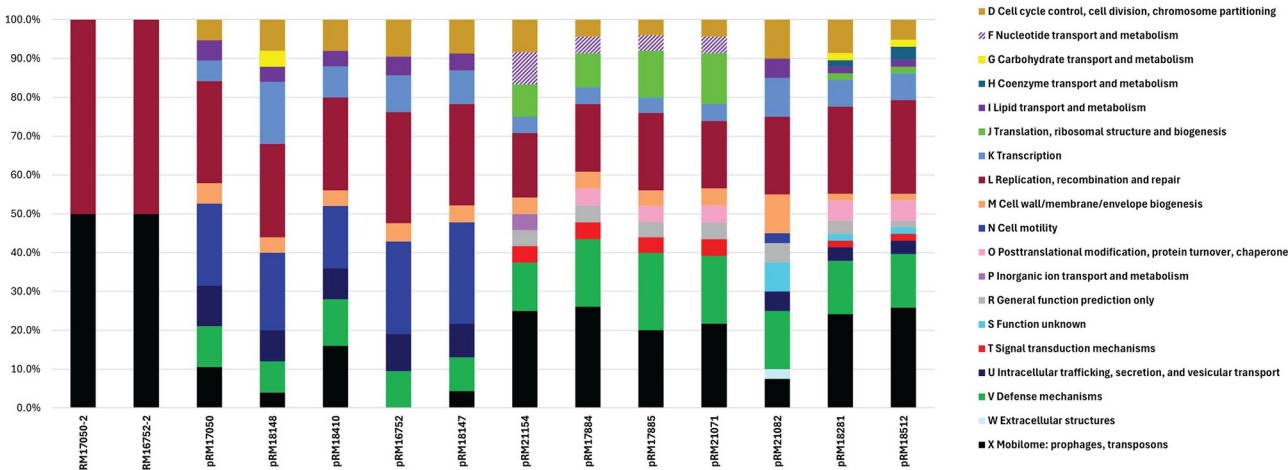

**Fig 2. Functional gene categories identified in the *S. enterica* Infantis plasmids.** The assignment to functional clusters of orthologous genes categories was determined from the gene cluster summary file using the Anvi'o workflow for the examined *S.* Infantis plasmids, grouped in various clusters.

and recombination (L category) as well as defense mechanisms and transcription (V and K categories, respectively).

Subsequent analysis of cell cycle control and division functional category revealed the presence in the large plasmids of *parA* and *paB*-like genes, involved in plasmid division. Examination of genes involved in replication, recombination and repair demonstrated the large plasmids contained a site-specific recombinase *xerD* like gene. The phage-containing plasmids in cluster III and IncF plasmids in cluster II (p91487, p116460, and p126613) contained an additional site-specific recombinase like gene resembling *pinE*. These phage-containing plasmids also harbor a DNA methylase-like gene and a gene resembling a DNA recombination-dependent growth factor RdgC. The remaining plasmids had a different DNA methylase *yhdj* identified, as well as genes supporting DNA repair.

To characterize genes associated with defense mechanisms, cluster III plasmids were found to contain *higA* and *higB* like genes, encoding the HigAB toxin antitoxin module for plasmid maintenance. The IncI1 plasmids, including pRM18148, contained the *ardA* antirestriction gene. The IncF plasmids contained a type I restriction system as well as an *emrAB* coding for an efflux pump system.

On the subsequent analysis of genes involved in transcription, all large plasmids contained a form of a SOS-response transcriptional repressor LexA. In particular, plasmids pRM18512 and pRM18281 contained a predicted transcriptional regulator with a different ribbon-helix-helix (RHH_1) domain and predicted DNA-binding transcriptional regulator. Plasmid pRM21082 contained a predicted RNA polymerase-binding transcription factor DksA, and a putative transcriptional regulator GlxA. Most IncI1 plasmids, with the exception of pRM18148, contained a predicted putA1 ribbon-helix-helix transcription regulator while plasmid pRM18148 alternately contained predicted DNA-binding proteins in the AraC and AcrA family. On the analysis of cell motility, the IncI1 plasmids contained the effector genes, *pilK-pilV* associated with the type II secretion system, including a pseudopilin, resembling a flagellum. The IncF plasmids contained the RNA binding protein ProQ, involved in post-translational regulation.

Examination of the mobilome functional category X, including prophages and transposons, showed that cluster III plasmids contained genes involved in regulation and structure of the phage. Additional observations revealed the presence of genes with a lytic trans-glycosylase *lysM* domain, assigned to biogenesis of the cell wall and membrane category M (Fig 2). Also, functional genes associated with signal transduction mechanisms (category T) included the *cpdA* gene coding for 3',5'-cyclic AMP phosphodiesterase in the cluster III plasmids and the sRNA binding gene *proQ* in the cluster V IncF plasmids. Moreover, the IncF plasmids pRM18512 and pRM18281 contained a transposase system in the mobilome functional category that encoded microcin V (*mccV*) as well as a predicted arabinose efflux permease *araJ* like gene (category G) (Fig 2). The IncF plasmids also contained the macrolide efflux protein A MefA as part of a major facilitator superfamily involved in substrate transport across membranes. Only the IncF plasmid pRM21082 harbored a pilus assembly gene *fimC* encoding for a chaperone identified in the functional category assigned to extracellular structures (category W).

## Tetracycline resistance determinants are present in the *S.* Infantis plasmid pRM18148

By using a validated database to screen for antimicrobial resistance genes, the analysis revealed that only one of the examined Infantis plasmids, the IncI plasmid pRM18148 had genes conferring tetracycline resistance (Fig 3). Phenotypic assays employing the disk diffusion method corroborated in strain RM18148 a strong resistance based on the absence of a zone of

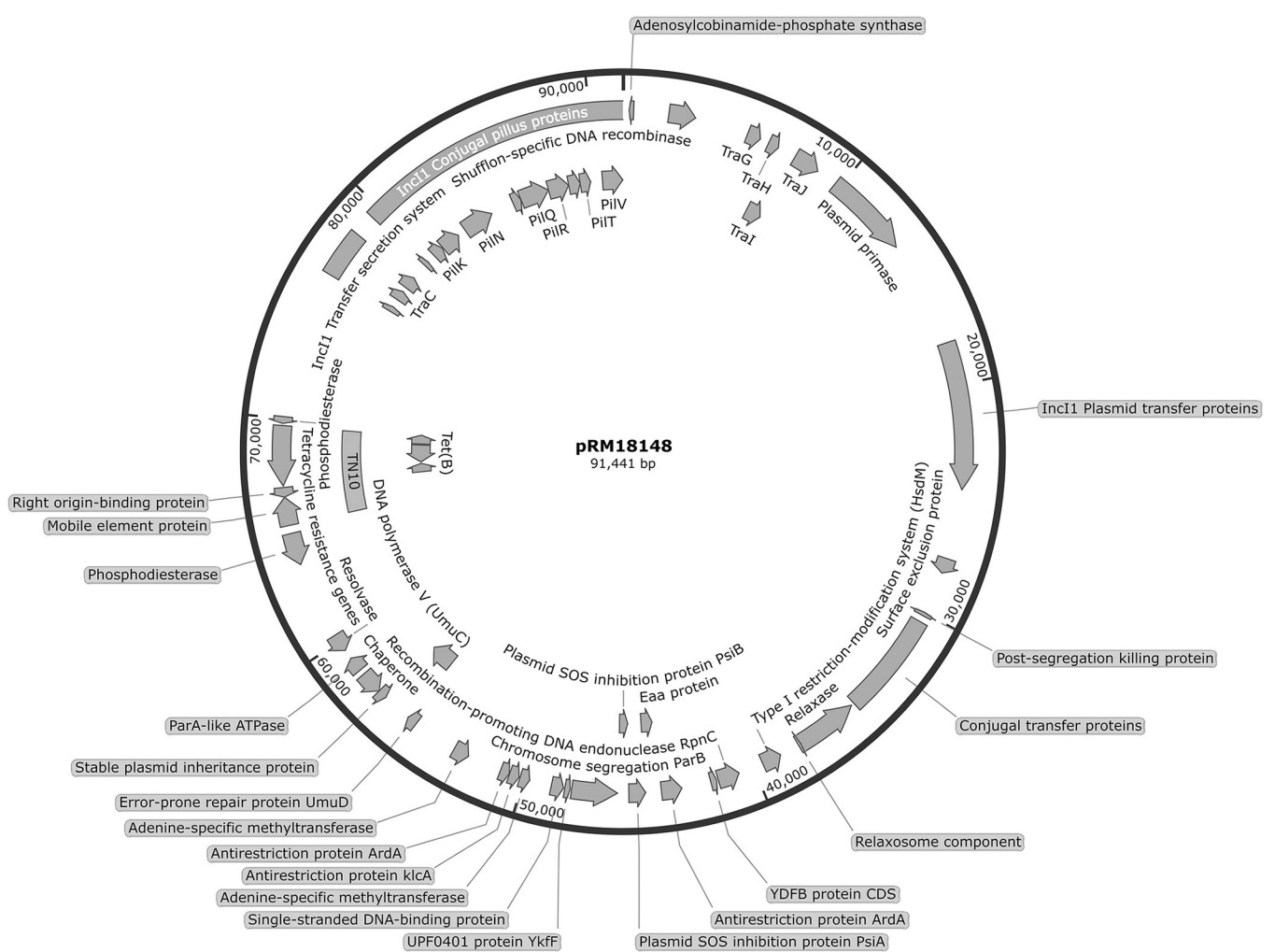

**Fig 3. Schematic diagram of *S.* Infantis plasmid pRM18148 conferring tetracycline resistance.** The RAST annotated nucleotide sequences for plasmid pRM18148 were uploaded to SnapGene software, version 7.2 (GSL Biotech LLC, San Diego, CA) to generate the plasmid map.

inhibition surrounding the tetracycline-infused disk. Further analysis with the TnCentral web portal of the region conferring tetracycline resistance in the Infantis plasmid pRM18148 indicated that these antimicrobial resistance genes were contained within a partial sequence of the Tn10 transposon (Fig 3), a composite mobile element which was also previously found in an unnamed plasmid identified in *S. enterica* serovar Heidelberg strain FDAARGOS_319 (Accession No. CP027411). Analysis of the coding sequences in the Infantis plasmid pRM18148 confirmed the presence of the TetB efflux pump, flanked by the tetracycline resistance transcriptional repressor TetR, the tetracyline resistance-associated transcriptional repressor TetC, and the multidrug resistant efflux pump AcrAB transcriptional activator RobA.

One difference identified in the Infantis plasmid pRM18148 was the presence of a putative adenosylcobinamide-phosphate synthase (*cbiB*) gene (Fig 3), a region with a potential role in conjugation and coenzyme $B_{12}$ biosynthesis. Other notable features identified in the Infantis plasmid pRM18148 included the SOS-inhibitors PsiA and PsiB, a shufflon-specific DNA recombinase, and the antirestriction protein ArdA (Fig 3). On the identification of the conjugal apparatus, plasmid pRM18148 was found to harbor the *traABC* region as well as the

relaxase protein NikB. Other notable genes needed for conjugation were a shufflon-specific DNA recombinase, and the *ardA* gene, and a *pil* gene cluster required for Type IV pilus formation.

## Discussion

*S. enterica* serovar Infantis is a clinically relevant serovar, and recent epidemiological studies have indicated a 167% increase in human infections in the United States [6, 8]. The unique characteristics of *S.* Infantis as a critical human pathogen is strongly correlated with the presence of plasmid pESI, providing resistance to multiple antibiotic classes as well as an enhanced biofilm formation, adhesion and invasion into host mammalian cells [11]. To further characterize the potential emergence of *S.* Infantis in agricultural regions for fresh produce in Central California, the objective of the present study was to determine the occurrence of plasmids, potentially conferring multidrug resistance in serovar Infantis strains, previously recovered from a survey of surface waters [27]. Although the analysis of the plasmid carriage failed to identify the multidrug resistant plasmid pESI in the recovered the *S.* Infantis strains from surface waters, the comparative sequence analysis of the detected plasmids with critically important conjugation groups IncI1 and IncF. In particular, IncI1 plasmids are a common plasmid type among the *Enterobacteriaceae* and have been isolated from food and clinical sources. Because of their potential to carry antibiotic resistance and virulence genes, IncI1 plasmids are a cause of public health concern [46]. Studies with *E. coli* and *Salmonella* 4,5,12:i:– strains showed that carriage of the IncI1 plasmid carried no measurable fitness cost on the bacteria in terms of association and invasion *in vitro* [46, 47]. IncF plasmids are also known for transferring antibiotic resistance determinants. The IncF family may contain SpvB, an ADP-ribosylating toxin, which allows it to cause systemic infection in humans. To date, the SvpB-IncF plasmid has been found in nine different *Salmonella* serovars, not including Infantis [48–50]. The pESI plasmid is a member of the IncF1B family [12].

Sequence comparison of the examined Infantis plasmids with those deposited in the databases indicated that the plasmids from this study showed significant sequence similarity to plasmids previously identified both locally and across the globe and indicated the ability of these plasmid types to move among the microbial population. In particular, the cluster I plasmids showed the closest similarity to a plasmid sequenced from a multidrug resistant *S. enterica* serovar Typhimurium strain 46 isolated from a clinical sample in China [51]. Interestingly, the cluster V plasmids were most similar to plasmid pRM11060-2, previously described from a different *Salmonella* strain with the serotype 6,8:d:– recovered from surface waters collected in Central California in 2009 [52]. When compared to plasmids grouped in cluster V, plasmid pRM11060-2 contained an additional 154-kb fragment encoding a prophage and hemolysin [52]. The *S.* Infantis strains RM18512 and RM18281 were isolated from water samples collected in 2013–2014 in the same agricultural region [27]. These observations suggested that the IncF plasmid, detected in Infantis and 6,8:d:– strains isolated 4–5 years apart, appeared to be prevalent in this surveyed agricultural region in Central California and are regularly re-introduced and move among different strains and serotypes of *Salmonella*. Consistent with this hypothesis, further analysis of the homogeneity indexes indicated that a high degree of gene variability was identified in relevant traits related to plasmid segregation, conjugation, replication, and stability. Variability in these among all the plasmid genes could potentially have an effect on the ability of these plasmids to move determinants among different genera of bacteria or different hosts within a genus in the environment, and this variability could subsequently influence the bacterial host virulence, adaptation and fitness to environments [16, 17, 50]. The closest similarities of the IncI1 plasmid pRM18148 were reported in *Salmonella* serovars

Kentucky and the monophasic 4, [5],12:i:-. The Infantis plasmid carried the *cdiB* gene, which encodes an integral membrane protein important for the last step in Vitamin B$_{12}$ synthesis in *Salmonella* serovar Typhimurium [53].

Moreover, the present study also documented the presence of phage plasmids, a plasmid type with a potential for both horizontal and vertical transfers [54]. Phage plasmids have been described previously in *Salmonella* [54–56], and this type of mobilizable genetic element is capable of conferring extended-spectrum β-lactamase resistance in *S. enterica* serovar Typhi and other bacteria [54, 57]. In the present study, the phage plasmids in cluster III showed sequence similarity to a plasmid from the clinical *S. enterica* strain 2013K-1747. Although there is limited published information about location, transmission vehicle and illness caused by strain 2013K-1747, the genome sequence was included as part of a larger study of genomic typing schemes for *Salmonella* [58]. Subsequent genome sequence analysis, using SeqSero2 algorithm [36], revealed strain 2013K-1747 to be a *S. enterica* serovar Adalaide. The appearance of very similar phage plasmids in two serovars, Infantis and Adalaide, indicate movement of this type of plasmid among *Salmonella* strains.

The presence of cell cycle control genes is to be expected among plasmids that are maintained in populations, particularly those genes that control replication, recombination, and repair. The ParABS DNA partitioning system was originally described in low copy number plasmids, like those described here [59, 60]. Similarly, the large plasmids contained a gene similar to *xerD*, which is involved in the transfer of genetic information, specifically the resolution of plasmid multimers and integration of DNA elements into host genomes [61]. This system allows for an extension of the host range of the plasmids and may play a role in the movement of these plasmids between strains and between genera.

Although no virulence factors were initially detected according to the output provided by the Virulence Finder Database [42], subsequent analysis of the functional gene categories in all of the *S.* Infantis plasmids identified several pathogenesis determinants conferring *Salmonella* an ability to potentially adhere, invade as well as colonize and survive in the host mammalian cells. In particular, the HigAB toxin-antitoxin module, present in the Cluster III plasmids, is involved in plasmid maintenance. This marker can also alter the production of virulence factors, as previously shown for another human pathogen, *Pseudomonas aeruginosa* [62]. The toxin-antitoxin systems were first described in F-plasmids and work with toxin genes present on the chromosome to allow for the survival of cells only if they retain the plasmid encoding the cognate antitoxin. These systems have been described in several different species of bacteria, including human pathogenic bacteria, and can be involved in virulence, stress response, and protein secretion [63, 64]. Other pathogenesis determinants include the *ardA* gene present in the IncI1 plasmids with a function in evading host Type I DNA restriction systems, and allowing plasmid-borne virulence genes to be maintained and spread [65]. The *emrAB* efflux coding system, which was present in the IncF plasmids, plays roles in both antibiotic and chromate resistance, and has been found in other human pathogenic bacteria isolated from the environment [66, 67]. The *dksA* gene, which was present in the IncF plasmid pRM21082, plays a role in redox-based signaling in response to nitrosative and oxidative stress in *Salmonella*, and control of these defenses contributes to *Salmonella* pathogenesis [68]. The *pilK-pilV* genes, which were present in the IncI1 plasmids, are expressed in *Salmonella* Typhi and are thought to play a role in the virulence of that pathogen. In other *Salmonella* isolates, the *pil* genes on the IncI1 R64 plasmid encode the Type IVb pilus for conjugation [69]. Mutants in *pilS*, the major pilin subunit, had reduced adhesion and invasion of human epithelial gastrointestinal cells *in vitro* [70, 71].

Additional genes that may have roles in pathogenesis included *mccV* in the mobilome of the Cluster V, IncF plasmids, which encodes microcin V, a low molecular weight antibiotic

involved in competitive interactions among species belonging to *Enterobacteriaceae* and with a role in promoting a successful colonization and survival in host epithelial cells [72, 73]. These same plasmids also contained the arabinose efflux permease *araJ*, which was previously implicated as an inducer of biofilm production in *S.* Typhimurium [74]. Additionally, the presence of the macrolide efflux protein A MefA, a member of the major facilitator superfamily [75], may confer an increased ability for the uptake of nutrients and extrusion of harmful compounds such as antibiotics and heavy metals [75, 76]. Moreover, the expression of the ProQ RNA binding protein in these IncF plasmids regulates the expression of virulence gene repertoire, promoting *Salmonella* intracellular growth during host cell colonization [77].

In the analysis of functional gene categories with a role in the biogenesis of the cell wall and membrane with a role in virulence, the *S.* Infantis cluster III phage plasmids harbored genes with a lytic trans-glycosylase LysM domain, proposed to be involved in a wide array of functions such as the routine maintenance of the peptidoglycan structure, engaging in bacterial resistance against the action of β-lactam antibiotics, and virulence pathways during host invasion [78]. Genes implicated in signal transduction mechanisms in the cluster III plasmids included the expression of 3′,5′-cAMP phosphodiesterase, required for fine-tuning the intracellular levels of the global virulence regulator cAMP in bacterial pathogens [79].

The use of the Comprehensive Antibiotic Resistance Database [41] in the examined *S.* Infantis strains revealed that only the IncI1 plasmid pRM18148 carried an antibiotic resistance marker. Using the TnCentral web portal [80], markers conferring resistance to tetracycline were found within a partial sequence of the Tn10 transposon, and tetracyline resistance in strain RM18148 was corroborated phenotypically in this study. The Tn10 transposon is a well-characterized composite element with the ability to move between other plasmids and host genomes [81], and this composite mobile element has been previously documented in *S. enterica* serovar Heidelberg strain FDAARGOS_319 (Accession No. CP027411) [82]. Tetracyclines are widely used agents in agricultural settings [83], and resistance to these agents has been one of the most commonly identified in the plasmids among *S. enterica* isolates from food animals in the United States [84], highlighting the need for better monitoring on the use of these agents in agriculture. Additionally, pRM18148 is an IncI1 plasmid and carries the necessary coding for conjugation, indicating that the tetracycline resistance cassette could readily move between *Salmonella* strains in the environment [46]. The *traABC* region and the *nikB* gene encode proteins needed for the formation of the relaxosome to initiate transfer of plasmid DNA [46]. The antirestriction protein ArdA functions in bacterial conjugation to allow an unmodified plasmid to evade restriction modification system in the recipient host [65, 85].

In summary, the present study employed high-resolution sequencing to characterize mobilizable plasmids, belonging to different incompatibility groups, in *S. enterica* serovar Infantis strains, recovered from surface waters adjacent to leafy greens farms in California. The comparative analysis of functional genes led to the identification of pathogenesis and antimicrobial resistance determinants that can contribute to the virulence potential of the *S.* Infantis strains from an important agricultural region.

## Conclusions

To further characterize plasmid carriage, the objective of the present study was to examine the plasmidome among strains of *S. enterica* serovar Infantis, recovered from surface waters proximal to leafy greens fields in Central California, an important agricultural region in the United States. Among the *Salmonella* serovars, Infantis continues to contribute significantly to the incidence of human infections worldwide due to the presence of the megaplasmid pESI. Although the plasmid pESI was not identified in the examined *S.* Infantis in this study,

mobilizable IncI1, IncF, and phage plasmids were detected that were harboring determinants implicated in virulence and antimicrobial resistance. The analysis of functional gene categories of the annotated sequences in the Infantis plasmids revealed determinants with roles implicated in promoting the successful colonization and survival of host by enhancing the adhesion and invasion of epithelial cells and by increasing the nutrient uptake and the extrusion of harmful compounds and antimicrobials. Additionally, the identification of the composite Tn10 transposon, conferring tetracycline resistance in strain RM18148, underline the necessity of better monitoring of commonly used agents in agriculture. Moreover, the findings from this study provided evidence for IncF plasmid movement among different *Salmonella* serovars, recovered from surface waters in Central California. The high similarity of the Infantis IncF plasmids to plasmid pRM11060-2 from a different *Salmonella* with the serotype 6,8:d:−, isolated in a previous surface water survey, demonstrated that these plasmids continue to be prevalent among strains and serovars of *S. enterica* in this agricultural region for leafy greens. This study has provided fundamental information based on annotated sequence data, and additional comparative and functional analyses are thus needed to determine the expression patterns of the identified genes and dissect how their expression contributes to the virulence phenotypes. Leafy greens have been previously a vehicle of foodborne illness, and surface waters can be a potential a route of transmission of bacterial pathogens; therefore, subsequent phenotypical studies are aimed at further categorizing the virulence potential as well as the environmental fitness, adaptability and persistence of the examined *S.* Infantis strains, recovered from this important agricultural region for fresh produce.

## Supporting information

**S1 Table. *Salmonella enterica* serovar Infantis strains screened for plasmids in this study.** (DOCX)

## Acknowledgments

The authors thank Dr. Dayna Harhay (USDA-ARS, U.S. Meat Animal Research Center, Clay Center, NE) for providing technical advice and *E. coli* strains harboring control plasmids.

## Author Contributions

**Conceptualization:** Beatriz Quiñones, Lisa Gorski.

**Data curation:** Bertram G. Lee, Ashley Avilés Noriega, Lisa Gorski.

**Formal analysis:** Beatriz Quiñones, Bertram G. Lee, Lisa Gorski.

**Funding acquisition:** Beatriz Quiñones, Lisa Gorski.

**Investigation:** Bertram G. Lee, Ashley Avilés Noriega.

**Methodology:** Bertram G. Lee, Ashley Avilés Noriega.

**Project administration:** Beatriz Quiñones, Lisa Gorski.

**Resources:** Beatriz Quiñones, Lisa Gorski.

**Software:** Bertram G. Lee.

**Supervision:** Beatriz Quiñones, Lisa Gorski.

**Validation:** Beatriz Quiñones, Bertram G. Lee, Ashley Avilés Noriega, Lisa Gorski.

**Visualization:** Beatriz Quiñones, Bertram G. Lee, Ashley Avilés Noriega, Lisa Gorski.

**Writing – original draft:** Beatriz Quiñones, Bertram G. Lee, Ashley Avilés Noriega, Lisa Gorski.

**Writing – review & editing:** Beatriz Quiñones, Bertram G. Lee, Lisa Gorski.

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
