## [Decision Letter · Decision Letter 0]

24 Oct 2024

PONE-D-24-42911Plasmidome of Salmonella enterica serovar Infantis recovered from surface waters in a major agricultural region for leafy greens in CaliforniaPLOS ONE

Dear Dr. Gorski,

Thank you for submitting your manuscript to PLOS ONE. After careful consideration, we feel that it has merit but does not fully meet PLOS ONE’s publication criteria as it currently stands. Therefore, we invite you to submit a revised version of the manuscript that addresses the points raised during the review process.

Please submit your revised manuscript by Dec 08 2024 11:59PM. If you will need more time than this to complete your revisions, please reply to this message or contact the journal office at plosone@plos.org. Please include the following items when submitting your revised manuscript:A rebuttal letter that responds to each point raised by the academic editor and reviewer(s). You should upload this letter as a separate file labeled 'Response to Reviewers'.A marked-up copy of your manuscript that highlights changes made to the original version. You should upload this as a separate file labeled 'Revised Manuscript with Track Changes'.An unmarked version of your revised paper without tracked changes. You should upload this as a separate file labeled 'Manuscript'.If applicable, we recommend that you deposit your laboratory protocols in protocols.io to enhance the reproducibility of your results. Protocols.io assigns your protocol its own identifier (DOI) so that it can be cited independently in the future. For instructions see: https://journals.plos.org/plosone/s/submission-guidelines#loc-laboratory-protocols. Additionally, PLOS ONE offers an option for publishing peer-reviewed Lab Protocol articles, which describe protocols hosted on protocols.io. Read more information on sharing protocols at https://plos.org/protocols?utm_medium=editorial-email&utm_source=authorletters&utm_campaign=protocols.

We look forward to receiving your revised manuscript.

Kind regards,

Mabel Kamweli Aworh, DVM, MPH, PhD. FCVSN

Academic Editor

PLOS ONE

Journal Requirements:

3. Please note that your Data Availability Statement is currently missing the direct link to access each database (GenBank accession numbers PQ365737-PQ365750). If your manuscript is accepted for publication, you will be asked to provide these details on a very short timeline. We therefore suggest that you provide this information now, though we will not hold up the peer review process if you are unable.

**Additional Editor Comments:**

Thank you for submitting your manuscript. Please address the following revisions to improve the clarity and structure of your paper:

<ol><li>**Restructure the manuscript**:

Separate the Results and Discussion sections. The Discussion section should not have any subtitles. In the Discussion, avoid repeating the results; instead, provide an interpretation of your findings, discuss their implications, and compare them with similar studies in the literature. Please do not cite any figures or tables in the Discussion section.The Conclusion section should provide a summary of the key findings from your study without any citations. Please ensure the Conclusion is focused on summarizing the main results and offering recommendations based on your findings. The recommendations should be provided in the final paragraph of the Conclusion section.<li>**Limitations**:

Highlight the key limitations of your study as the final paragraph of the Discussion section.

Reviewers' comments:

Reviewer's Responses to Questions

**Comments to the Author**

1. Is the manuscript technically sound, and do the data support the conclusions?

Reviewer #1: Yes

Reviewer #2: Yes

2. Has the statistical analysis been performed appropriately and rigorously? 

Reviewer #1: Yes

Reviewer #2: Yes

3. Have the authors made all data underlying the findings in their manuscript fully available?

Reviewer #1: Yes

Reviewer #2: Yes

4. Is the manuscript presented in an intelligible fashion and written in standard English?

Reviewer #1: Yes

Reviewer #2: Yes

5. Review Comments to the Author

Reviewer #1: I will first like to congratulate the author for a well researched work he's put foward here, it is really rigorous. I have just few issues as outlined below.

Line 152: "Twelve strains were selected for sequence analysis." so, is it twelve strains out of the 34, 14 or the 20. please specify out of which group is the 12 selected from.

Lines 206-209: "As further demonstrated in the comparative analysis with the Anvi’o workflow (Fig 1), various homogeneity indexes were determined to measure the variation and frequency of their occurrence in the detected plasmid genes with an index of 1 being the highest value indicating no variation." please specify the indexes tool used for this study.

Did you have some limitation in this study? if yes please mention some.

Did you have ethical approval for this study? please if yes is better if you include it in.

what recommendation can you give based on the findings?

include the funding information for this study and possible conflict of interest if any.

Reviewer #2: Thank you for the opportunity to review this manuscript. After a thorough reading, I found the manuscript to be well-written, scientifically sound, and free of any significant issues. The research is well-structured, and the conclusions drawn are appropriate based on the presented data. I have no further suggestions or revisions at this time, and I believe this manuscript is ready for publication in its current form.

6. PLOS authors have the option to publish the peer review history of their article (what does this mean?). If published, this will include your full peer review and any attached files.

Reviewer #1: **Yes: **Umar Muhammad Umar

Reviewer #2: No

---

## [Author Response · Author response to Decision Letter 0]

13 Nov 2024

RESPONSES TO EDITOR’S COMMENTS| MANUSCRIPT ID PONE-D-24-42911:

1. EDITOR’S COMMENT: Please ensure that your manuscript meets PLOS ONE's style requirements, including those for file naming. 

AUTHORS’ REPLY: The manuscript was formatted according to PLOS ONE’s guidelines.

2. EDITOR’S COMMENT: In your Methods section, please provide additional information regarding the permits you obtained for the work. Please ensure you have included the full name of the authority that approved the field site access and, if no permits were required, a brief statement explaining why.

AUTHORS’ REPLY: In the Material and Methods section (lines 77-80), additional statements were inserted in the revised manuscript to indicate that the present study did not require permits issued for since all of the sampling sites were located on public lands within an approximate 20-mile radius of Salinas, California (USA). No specific permissions were required for sampling on the public lands, and these studies did not involve sampling any endangered or protected species. 

3. EDITOR’S COMMENT: Please note that your Data Availability Statement is currently missing the direct link to access each database (GenBank accession numbers PQ365737-PQ365750).

AUTHORS’ REPLY: Given that the direct link to the sequences through the BioProject site is unavailable, the Data Availability Statement (lines 157-160) was edited in the revised manuscript by providing an alternate site (https://www.ncbi.nlm.nih.gov/nuccore/) to access the plasmid sequences in the NCBI nucleotide database.

4. EDITOR’S COMMENT: We note that you have included the phrase “data not shown” in your manuscript. Unfortunately, this does not meet our data sharing requirements. PLOS does not permit references to inaccessible data. We require that authors provide all relevant data within the paper, Supporting Information files, or in an acceptable, public repository. Please add a citation to support this phrase or upload the data that corresponds with these findings to a stable repository (such as Figshare or Dryad) and provide and URLs, DOIs, or accession numbers that may be used to access these data. Or, if the data are not a core part of the research being presented in your study, we ask that you remove the phrase that refers to these data.

AUTHORS’ REPLY: To address the Editor’s comment, the Material and Methods (lines 90-96) and Results (lines 325-329) sections are now included in the revised manuscript text on the analysis of the in vivo antimicrobial resistance against tetracycline, which was originally mentioned as “data not shown”. 

5. EDITOR’S COMMENT: Please review your reference list to ensure that it is complete and correct. 

AUTHORS’ REPLY: The reference list was reviewed to be complete and correct.

6. EDITOR’S COMMENT Restructure the manuscript. Separate the Results and Discussion sections. The Discussion section should not have any subtitles. In the Discussion, avoid repeating the results; instead, provide an interpretation of your findings, discuss their implications, and compare them with similar studies in the literature. Please do not cite any figures or tables in the Discussion section. The Conclusion section should provide a summary of the key findings from your study without any citations. Please ensure the Conclusion is focused on summarizing the main results and offering recommendations based on your findings. The recommendations should be provided in the final paragraph of the Conclusion section.

AUTHORS’ REPLY: The manuscript was extensively revised, and the Results (pages 9-18) and Discussion (pages 18-23) are now in separate sections. The highlights of the present study and recommendations for future analyses have been included in the revised Conclusions section (page 24). 

7. EDITOR’S COMMENT: Limitations: Highlight the key limitations of your study as the final paragraph of the Discussion section.

AUTHORS’ REPLY: The limitations of the current study were included after the Discussion section (page 24) by stating that the current study has provided fundamental information based on annotated sequence data. Given that only genomic data is provided in this study, additional comparative and functional analyses are thus needed to determine the expression patterns of the identified genes and dissect how their expression contributes to the virulence phenotypes.

RESPONSES TO REVIEWER #1’S COMMENTS| MANUSCRIPT ID PONE-D-24-42911:

1. REVIEWER’S COMMENT: Line 152: "Twelve strains were selected for sequence analysis." so, is it twelve strains out of the 34, 14 or the 20. please specify out of which group is the 12 selected from.

AUTHORS’ REPLY: An additional statement was inserted in the revised manuscript text (lines 166-167) to specify that the examined 12 Infantis strains were selected based on the high yield recovery of the detected large plasmids.

2. REVIEWER’S COMMENT: Lines 206-209: "As further demonstrated in the comparative analysis with the Anvi’o workflow (Fig 1), various homogeneity indexes were determined to measure the variation and frequency of their occurrence in the detected plasmid genes with an index of 1 being the highest value indicating no variation." please specify the indexes tool used for this study.

AUTHORS’ REPLY: To address the Reviewer #1’s question, the manuscript text was revised (lines 145-148) to indicate that the function in the Anvi’o workflow called “anvi-compute-gene-cluster-homogeneity” was employed to compute the geometric and functional homogeneity indexes, and these indexes provided measures of similarity in the detected plasmid genes with an index of 1 being the highest value to indicate no variation.

3. REVIEWER’S COMMENT: Did you have some limitation in this study? if yes please mention some.

AUTHORS’ REPLY: The limitations of the current study were included after the Discussion section (page 24) by stating that the current study has provided fundamental information based on annotated sequence data. Given that only genomic data is provided in this study, additional comparative and functional analyses are thus needed to determine the expression patterns of the identified genes and dissect how their expression contributes to the virulence phenotypes.

4. REVIEWER’S COMMENT: Did you have ethical approval for this study? please if yes is better if you include it in.

AUTHORS’ REPLY: In the Material and Methods section (lines 77-80), additional statements were inserted in the revised manuscript to indicate that the present study did not require permits issued for since all of the sampling sites were located on public lands within an approximate 20-mile radius of Salinas, California (USA). No specific permissions were required for sampling on the public lands, and these studies did not involve sampling any endangered or protected species

5. REVIEWER’S COMMENT: What recommendation can you give based on the findings?

AUTHORS’ REPLY: Recommendations on the directions of future studies based on the results obtained in the present study are included in the Conclusions section (page 24). Recent report have documented that leafy greens have been previously a vehicle of foodborne illness, and surface waters can be a potential a route of transmission of bacterial pathogens. Therefore, subsequent phenotypical studies are aimed at further categorizing the virulence potential as well as the environmental fitness, adaptability and persistence of the examined S. Infantis strains, recovered from this important agricultural region for fresh produce.

6. REVIEWER’S COMMENT: Include the funding information for this study and possible conflict of interest if any.

AUTHORS’ REPLY: As indicated in the author guidelines for Plos One, the funding information and conflict of interest needed to be uploaded separately from the manuscript text by using the journal submission site.

---

## [Editor Report · Decision Letter 1]

22 Nov 2024

PONE-D-24-42911R1Plasmidome of Salmonella enterica serovar Infantis recovered from surface waters in a major agricultural region for leafy greens in CaliforniaPLOS ONE

Dear Dr. Gorski,

Thank you for submitting your manuscript to PLOS ONE. After careful consideration, we feel that it has merit but does not fully meet PLOS ONE’s publication criteria as it currently stands. Therefore, we invite you to submit a revised version of the manuscript that addresses the points raised during the review process.

We look forward to receiving your revised manuscript.

Kind regards,

Mabel Kamweli Aworh, DVM, MPH, PhD. FCVSN

Academic Editor

PLOS ONE

Journal Requirements:

Additional Editor Comments :

Thank you for addressing the previous comments. While your revisions are appreciated, there are still areas that require improvement to align the manuscript with journal standards:

<ol><li>**Restructure the Manuscript**

In the *Discussion* section, avoid repeating the results. Focus on interpreting your findings, discussing their implications, and comparing them with existing studies in the literature.<li>**Results Section**

Remove all citations from the *Results* section. The *Results* should strictly state the findings, supported by accompanying tables and figures. Cite tables and figures as appropriate, but do not provide explanations or cite literature in this section.Specifically, citations found in lines 214 and 252–398 of the tracked manuscript version must be removed.<li>**Discussion Section**

Revise the *Discussion* section to include explanations, implications of findings, and comparisons with relevant literature. Do not restate the results in this section.<li>**Other Issues**

Delete the phrase "see Materials and Methods" from line 245 in the tracked manuscript.

Please make these revisions and resubmit the manuscript for further evaluation. Let us know if you have any questions or require additional clarification.

---

## [Author Response · Author response to Decision Letter 1]

9 Dec 2024

1. Restructure the Manuscript

o In the Discussion section, avoid repeating the results. Focus on interpreting your findings, discussing their implications, and comparing them with existing studies in the literature.

• The Results and Discussion sections were edited. Results were removed from the Discussion, references were removed from Results. 

2. Results Section

o Remove all citations from the Results section. The Results should strictly state the findings, supported by accompanying tables and figures. Cite tables and figures as appropriate, but do not provide explanations or cite literature in this section.

• Citations were removed from the Results section.

o Specifically, citations found in lines 214 and 252–398 of the tracked manuscript version must be removed.

• Citations were removed from the results section.

3. Discussion Section

o Revise the Discussion section to include explanations, implications of findings, and comparisons with relevant literature. Do not restate the results in this section.

• The Results and Discussion sections were edited. Results were removed from the Discussion, references were removed from Results. 

4. Other Issues

o Delete the phrase "see Materials and Methods" from line 245 in the tracked manuscript.

• This phrase was deleted.

---

## [Editor Report · Decision Letter 2]

12 Dec 2024

Plasmidome of Salmonella enterica serovar Infantis recovered from surface waters in a major agricultural region for leafy greens in California

PONE-D-24-42911R2

Dear Dr. Gorski,

We’re pleased to inform you that your manuscript has been judged scientifically suitable for publication and will be formally accepted for publication once it meets all outstanding technical requirements.

Kind regards,

Mabel Kamweli Aworh, DVM, MPH, PhD. FCVSN

Academic Editor

PLOS ONE
---

## [Editor Report · Acceptance letter]

13 Dec 2024

PONE-D-24-42911R2 

PLOS ONE

Dear Dr. Gorski, 

I'm pleased to inform you that your manuscript has been deemed suitable for publication in PLOS ONE. Congratulations! Your manuscript is now being handed over to our production team.

Kind regards, 

on behalf of

Dr. Mabel Kamweli Aworh 

Academic Editor

PLOS ONE